# Smectic phase in suspensions of gapped DNA duplexes

Miroslaw Salamonczyk[1,*], Jing Zhang[2,3,*], Giuseppe Portale[4], Chenhui Zhu[5], Emmanuel Kentzinger[6], James T. Gleeson[1], Antal Jakli[1], Cristiano De Michele[7], Jan K.G. Dhont[2,8], Samuel Sprunt[1] & Emmanuel Stiakakis[2]

Smectic ordering in aqueous solutions of monodisperse stiff double-stranded DNA fragments is known not to occur, despite the fact that these systems exhibit both chiral nematic and columnar mesophases. Here, we show, unambiguously, that a smectic-A type of phase is formed by increasing the DNA's flexibility through the introduction of an unpaired single-stranded DNA spacer in the middle of each duplex. This is unusual for a lyotropic system, where flexibility typically destabilizes the smectic phase. We also report on simulations suggesting that the gapped duplexes (resembling chain-sticks) attain a folded conformation in the smectic layers, and argue that this layer structure, which we designate as smectic-fA phase, is thermodynamically stabilized by both entropic and energetic contributions to the system's free energy. Our results demonstrate that DNA as a building block offers an exquisitely tunable means to engineer a potentially rich assortment of lyotropic liquid crystals.

[1] Department of Physics and Chemical Physics Interdisciplinary Program, Kent State University, Kent, Ohio 44242, USA. [2] Institute of Complex Systems ICS-3, JARA-SOFT, Forschungszentrum Jülich, Leo-Brandt-Str, Jülich D-52425, Germany. [3] Department of Environmental Nano-materials, Research Center for Eco-Environmental Sciences, Chinese Academy of Sciences, Beijing 100085, China. [4] Zernike Institute for Advanced Materials, Department of Macromolecular Chemistry and New Polymeric Materials, University of Groningen, Nijenborgh 4, 9747 AG Groningen, The Netherlands. [5] Advanced Light Source, Lawrence Berkeley National Laboratory, Berkeley, 94720 California, USA. [6] Jülich Centre for Neutron Science JCNS and Peter Grünberg Institut PGI, JARA-FIT, Forschungszentrum Jülich, Jülich D-52425, Germany. [7] Department of Physics, Sapienza Università di Roma, Piazzale A. Moro 5, Roma 00185, Italy. [8] Department of Physics, Heinrich-Heine-Universität Düsseldorf, Universitätsstrae 1, D-40225 Düsseldorf, Germany. * These authors contributed equally to this work. Correspondence and requests for materials should be addressed to C.D.M. (email: cristiano.demichele@roma1.infn.it) or to E.S. (email: e.stiakakis@fz-juelich.de).

Ordered lyotropic phases of densely packed DNA *in vivo* and *in vitro* share many similarities[1,2], so that the physics that underlies the phase behaviour of DNA[3] is of fundamental biological importance[4,5]. Phase transitions of double-stranded B-form DNA (dsDNA) in aqueous saline solutions have been extensively studied in the past, revealing a series of multiple lyotropic liquid crystal (LC) ordered phases at sufficiently high concentrations, depending mainly on the length of the dsDNA molecules and the sample preparation method[2,6–11].

The stability of these phases can be partly understood in terms of entropy-driven ordering of repulsive rigid or semi-flexible rod-shaped polymers to minimize the macromolecular excluded volume. The conceptual framework for the entropy-driven phase transition of solutions of monodisperse repulsive thin hard rods of length $L$ and diameter $D$ from an isotropic ($I$, orientationally disordered) fluid phase to a nematic ($N$, orientationally ordered) phase has been provided in the seminal work of Onsager[12]. Onsager[12] showed that for very long and thin rods (large $L/D$), translational entropy can be gained at the expense of orientational entropy beyond a volume fraction of $\varphi > \varphi_{I-N} = 3.29\, D/L$. Although the work by Onsager[12] was focused on the limit of very thin and very long rods ($L/D \gg 1$), computer simulations[13] on lyotropic hard repulsive spherocylinders show that the same entropy-driven LC ordering transitions occur for rod-like molecules with aspect ratios down to $L/D \approx 4.7$, and with the transition concentrations deviating from the prediction by Onsager[12] by amounts that depend on the value of $L/D$.

While these simulations suggest the absence of any kind of LC phase for $L/D < 4.7$, recent pioneering work on concentrated aqueous solutions of ultrashort blunt-ended dsDNA fragments with aspect ratios much $< 4.7$ (refs 14,15) revealed the formation of a nematic phase above a critical concentration. This unexpected finding was attributed to an attractive stacking interaction between the terminal ends of dsDNA, which induces the formation of linear aggregates which are long and rigid enough to form a chiral nematic (cholesteric) LC[16].

LC phases of slightly polydisperse rigid dsDNA fragments[17] with a length corresponding to $N_{bp} \sim 146 \pm 12$ (with $N_{bp}$ the number of base pairs), which is comparable to their persistence length ($l_p^{dsDNA} \sim 50$ nm–150 bp (ref. 18)) and for which $L/D \sim 25$, have been extensively studied mainly by polarized optical microscopy (POM)[2,6,11,19], X-ray scattering[6,7,20], nuclear magnetic resonance spectroscopy[17,21,22] and freeze-fracture electron microscopy methods[6]. With increasing DNA concentration, the following DNA mesophase transitions were identified: isotropic ($I$) to chiral nematic or cholesteric ($N^*$) to columnar hexagonal ($Col$) and finally to orthorhombic crystal ($K$). Interestingly in the above cascade of LC phases, no smectic ordering has been observed—notably absent is the most common smectic mesophase, the smectic-A ($Sm$-A) phase, in which two-dimensional (2D), fluid layers of molecules are stacked along the third dimension, which is also the axis (termed *director*) of orientational (nematic) order. Fragments of dsDNA around the above mentioned length, despite their polydispersity (ratio of the weight-averaged molecular weight to the number-averaged molecular weight, $M_w/M_n \sim 1.07$ (ref. 17), flexibility ($L/l_p \sim 1$) and electrostatic interactions, proved to fulfil the Onsager prediction for the isotropic-nematic phase transition by properly rescaling the effective duplex diameter to take the repulsive interactions into account[17,23]. In rod-like hard-core systems that are much stiffer than dsDNA, such as viral rod-like particles[24] and colloidal silica rods[25], a phase transition from nematic to the $Sm$-A phase is observed at sufficiently high concentration.

The absence of a $Sm$-A phase in DNA was elegantly demonstrated by Livolant *et al.*[6,26], who showed that 2D columnar positional ordering preempts the potential formation

of a one-dimensional layered smectic phase. Even though this behaviour is not fully understood, it could be explained on the basis of strand flexibility[27] or length polydispersity[28], both favouring the columnar phase over the smectic phase.

Here, we report conclusive small-angle X-ray scattering (SAXS) evidence, as well as computer simulations, that reveal it is possible to form a smectic phase in suspensions of short dsDNA fragments by introducing a flexible single-stranded DNA (ssDNA) region in the middle of the duplex. The stabilization of the lyotropic smectic phase by introducing a flexible spacer is not obvious and somewhat counter intuitive, since one would expect that a significant decrease in the system's stiffness will destabilize the smectic phase[29]. On the basis of a combination of physical arguments and our simulation results, we propose a specific model for the smectic layer structure in which the gapped duplexes predominantly adopt a folded configuration, with the rigid parts of our DNA-based chain-sticks lie side by side. We designate this novel smectic-A type of phase as a 'smectic-fA' phase, where 'f' stands for 'folded'.

## Results

**Synthesis**. In our synthetic approach we exploit the large difference in the persistence length between dsDNA ($\sim 50$ nm) and ssDNA ($l_p^{ssDNA} \sim 2$ nm (ref. 30)) to fabricate DNA duplexes possessing a central flexible region which is tunable in length (see the cartoons in Fig. 1a,b). These DNA duplexes thus consist of two stiff dsDNA fragments which are connected by a flexible ssDNA strand, resembling chain-stick like molecules. The main advantage of our synthetic scheme is that strictly monodisperse gapped DNA duplexes can be produced by the self-assembly of three partially complementary synthetic ssDNA strands in a 1:1:1 stoichiometric ratio, through a standard thermal annealing protocol (more details are given in the 'Methods' section and Supplementary Method 1). In particular, the length and the position of the paired ($L_{dsDNA}$) and unpaired ($L_{ssDNA}$) bases region (see Fig. 1a,b) can be controlled with sub-nanometre precision, at the level of a single base.

The systems involved in this study are two gapped duplexes (G-duplex) with a fixed length of the stiff dsDNA parts, $L_{dsDNA} = 48$ bp $\sim 16$ nm (using 0.33 nm per bp), and two lengths $L_{dsDNA}$ of the ssDNA flexible spacer, corresponding to 1 and 20 thymine (T) bases. We will refer to these two G-duplexes as the $G_{1T}$-duplex and the $G_{20T}$-duplex, respectively. As a reference system, we used the fully paired duplex counterpart of the $G_{20T}$-duplex (the F-duplex, depicted in Fig. 1a). Their gel-electrophoretic mobilities are presented in Fig. 1c. The poly-thymine sequence was selected for the ssDNA gap region due to the lack of propensity to form secondary structures.

The concentration dependence of the self-assembly behaviour of the proposed DNA duplexes in aqueous saline solutions was investigated using synchrotron and in-house SAXS and POM measurements. All experiments reported here were performed at a room temperature. Computer simulations were also carried out, and are discussed in the 'Monte Carlo Simulations' section below.

**SAXS from the F-duplex and G-duplexes**. While the F-duplex exhibits the isotropic to chiral nematic ($I/N^*$) transition, as expected for aqueous solutions of monodisperse dsDNA fragments[17,23,31], no evidence of a smectic phase is observed with increasing concentration. The isotropic to biphasic transition concentration is found to be 195 mg ml$^{-1}$ of DNA, which is in line with Onsager's prediction. An accurate theoretical estimate for such a transition most likely requires properly accounting for the duplex flexibility[32].

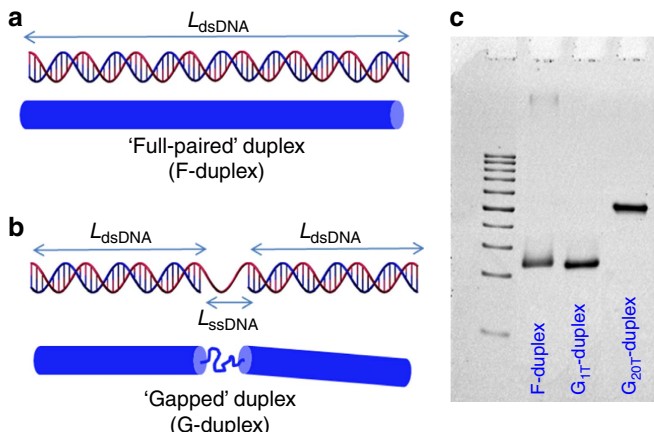

**Figure 1 | Synthesis and characterization of full-paired and gapped DNA duplexes.** Schematic representation of the DNA duplexes used for the exploration of the LC behaviour of rod-shaped molecules with tunable intrinsic flexibility. (**a**) dsDNA fragment formed by combining two complementary ssDNA strands (red and blue strands) with a length of $L_{dsDNA} = 116$ bp $\sim 38.3$ nm that is smaller than the dsDNA persistence length $l_p^{ssDNA} = 50$ nm. This molecule is referred to as the F-duplex, and is a model for a stiff rod as depicted in blue. (**b**) Three partially complementary ssDNA strands that form a DNA G-duplex. The red ssDNA strand has a length equal to the one used for the construction of the F-duplex in **a**. Base-pairing with the two shorter blue ssDNA strands results in a gapped G-duplex. The G-duplex thus consists of a central unpaired flexible ssDNA region with either 1 or 20 unpaired thymine bases, with a stiff dsDNA part with $L_{dsDNA} = 48$ bp attached to both sides of the flexible ssDNA part. The spacer with 20 unpaired thymine bases has a length $L_{ssDNA}$ that is much larger than the persistence length $l_p^{ssDNA} = 2$ nm of the corresponding ssDNA. (**c**) In all, 10% PAGE. From left to right: 50 bp DNA Ladder (bottom to the top: from 50 to 500 bp with a 50 bp step), F-duplex (with $L_{dsDNA} = 116$ bp), the $G_{1T}$-duplex, and the $G_{20T}$-duplex.

One-dimensional (1D)-SAXS profiles at room temperature for the F-duplex are shown in Fig. 2a for various concentrations. These profiles are obtained by azimuthally averaged radial scattering intensity of a 2D-SAXS scattering pattern, an example of which is shown in Fig. 2c.

For concentrations well above the $I/N^*$ transition, the 1D-SAXS profile (Fig. 2a, first two panels from the top, 300 mg ml$^{-1}$ and 287.4 mg ml$^{-1}$) reveals a single intense and narrow X-ray Bragg reflection, superimposed on a much broader peak, with the maximum of the scattering intensity located approximately at a scattering wave vector $q = q_{DNA} = 2.3$ nm$^{-1}$. The value of $q_{DNA}$ decreases slightly as the concentration is decreased (see Fig. 2a, second panel from the top, 287.4 mg ml$^{-1}$). At even lower concentrations (below about 260 mg ml$^{-1}$), the sharp peak disappears, and only the broad peak remains (see the middle scattering pattern in Fig. 2a). The latter originates from a liquid-like positional order between neighbouring, parallel DNA helices[7]. Assuming a local hexagonal packing, its central wavenumber corresponds to an interaxial distance between helices of $d = 4\pi/\sqrt{3}q = 3.64$ nm for the concentration of 247.2 mg ml$^{-1}$.

The appearance of the much sharper $q_{DNA}$-peak for higher concentrations marks a discontinuous transition from the $N^*$ to a more ordered state. This transition is most probably associated with a hexagonal-columnar ordering, similar to the one observed in suspensions of slightly polydisperse dsDNA fragments of similar length $L_{dsDNA} \sim 146$ bp in ref. 7.

Two-phase coexistence develops at the transition from the $I$ to the $N^*$ phase, as demonstrated in the two bottom scattering profiles in Fig. 2a. These are obtained by measuring at two

different locations within a sample that is in phase coexistence. The lower scattering curve is taken from the $N^*$ phase and the upper profile from the coexisting $I$ phase. Coexistence of the two phases is also evidenced by the depolarized images given in the insets of Fig. 2a.

Removing 20 bases from the central part of the DNA-double helix in the F-duplex, which yields the more flexible $G_{20T}$-duplex, results in very different phase behaviour for similar DNA concentrations, as can be seen from the scattering patterns in Fig. 2b. The images in Fig. 2d, taken through crossed polarizers, reveal an isotropic-nematic coexistence region in $G_{20T}$-duplex solutions at relatively low concentrations, similar to the case for the F-duplex, with the expected linear changes of the relative volumes of the two phases as a function of the overall $G_{20T}$-duplex concentration. The position and width of the higher $q$ peaks for the $G_{20T}$-duplex peaks (sharp peak at $q_{DNA}$ and broad peak) demonstrate the same concentration dependence (see Fig. 2b) as for the F-duplex.

Strikingly different, however, is the appearance of small-angle ($q < 1.0$ nm$^{-1}$) scattering peaks for $G_{20T}$-duplex concentrations in the range 231.8–300 mg ml$^{-1}$, as can be seen from the three top panels in Fig. 2b. A sharp principal scattering peak at a wave vector $q^*$ and several higher-order reflections appear, with wave vector ratios $q/q^*$ of 1:2:3:4. Such higher-order reflections are reminiscent of a lamellar structure. The position of the primary peak $q^*$ corresponds to a layered structure with a spacing between adjacent layers of $d = 2\pi/q^* = 34$ nm. There is a weak concentration dependence of the layer spacing, as can be seen from the three top scattering patterns in Fig. 2b: the spacing increases from 33.4 to 35.7 nm on increasing the concentration from 231.8 to 291.2 mg ml$^{-1}$.

The type of smectic phase can be determined by observing 2D-SAXS patterns of a shear-aligned sample, an example of which is given in Fig. 2e. Here the shear was due to flow along the capillary axis during sample loading. The peaks originating from correlations in $G_{20T}$-duplex length (arcs close to the beam stop) are oriented exactly perpendicularly to the peaks originating from correlations in duplex diameter (the outer broad arc, which corresponds to the high-$q$ peak in the 1D-SAXS profile). This clearly implies that the system self-organized in a $Sm$-A type of mesophase, in which the $G_{20T}$-duplex molecules within the layers are oriented parallel to the layer normal.

Alignment can also be achieved by the application of a magnetic field. DNA fragments tend to align perpendicular to the magnetic field[33]. In LC phases this results in an orientation of the director perpendicular to this field[34]. For a $Sm$-A type of phase in the $G_{20T}$-duplex, one would therefore expect to see scattering peaks corresponding to layering along the directions perpendicular to the magnetic field. The 2D-SAXS image in Fig. 2g, taken after aligning a 255 mg ml$^{-1}$ $G_{20T}$ sample in a 14 Tesla field for 48 h, clearly confirms this expectation.

Next, to emphasize the role of the flexibility introduced into the duplex by the ssDNA spacer on stabilizing the smectic phase, we performed SAXS measurements on concentrated solutions of $G_{1T}$-duplexes—that is, the G-duplex with a spacer of just a single thymine base. The blue dotted SAXS profile in Fig. 2b (second from top panel) was obtained on a solution of $G_{1T}$-duplex with essentially the same DNA concentration ($\approx 240$ mg ml$^{-1}$) as for the data on the solution of $G_{20T}$-duplex shown in the same panel. There is no evidence of smectic layering in the $G_{1T}$ sample (see also the SAXS image in Fig. 2f showing only a broad isotropic ring corresponding to the liquid-like ordering of neighbouring duplexes); this remains the case for concentrations up to 300 mg ml$^{-1}$ (see Supplementary Fig. 1 and Supplementary Note 1). The $Sm$-A phase is thus formed only when there is sufficient flexibility between the two rod-like dsDNA segments of gapped DNA duplex.

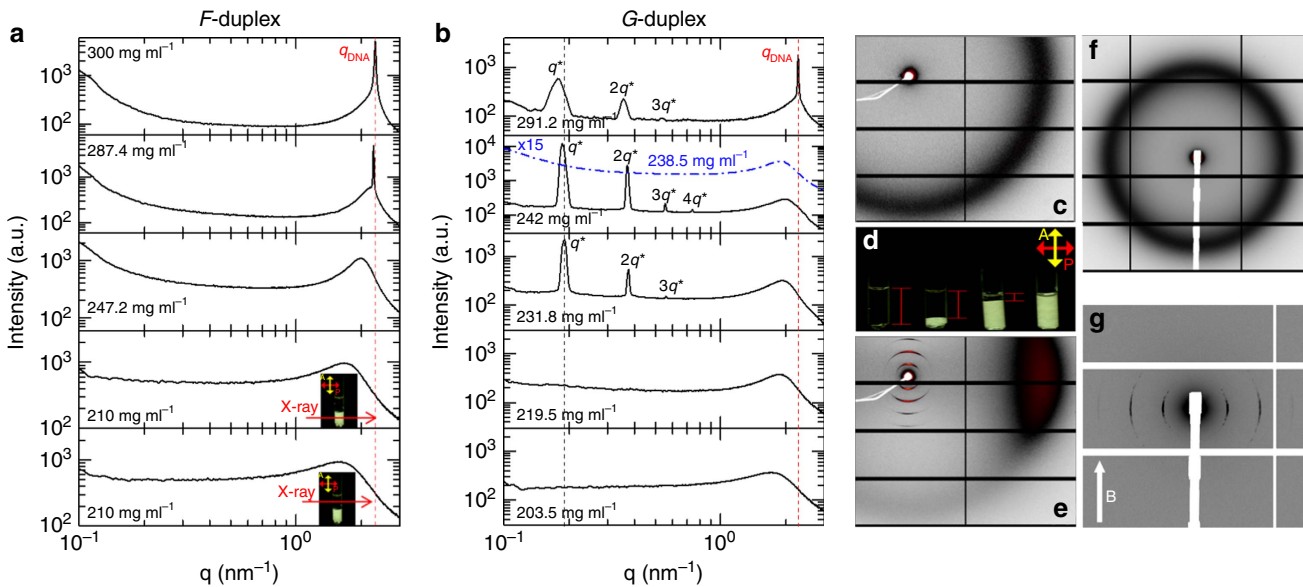

**Figure 2 | Sm-A ordering in a suspension of gapped DNA duplexes.** (**a**) X-ray scattering patterns for the F-duplex, with concentrations from top to the bottom are 300.0, 287.4, 247.2 and 210 mg ml$^{-1}$ (for the two lower patterns). The insets for the two lower concentrations show the location of the X-ray beam (the red arrows) through the bottom birefringent phase and the upper non-birefringent phase, respectively. The two bottom 1D-SAXS profiles are measured using an in-house SAXS setup. (**b**) X-ray scattering patterns for the $G_{20T}$-duplex, with concentrations from top to the bottom 291.2, 242.0, 231.8, 219.5 mg ml$^{-1}$ (taken from the LC region of the third capillary from the left in the photograph in **d**), and 203.5 mg ml$^{-1}$ (taken from the isotropic region of the second capillary from the left in the photograph of **d**). The correlation peaks assigned as $q^*$ together with their higher-order reflections at $2q^*$, $3q^*$, $4q^*$ correspond to a lamellar morphology. The blue-dashed curve in the second upper scattering pattern is for the $G_{1T}$-duplex at a concentration of 238.5 mg ml$^{-1}$ (the intensity is shifted by a factor of 15 for clarity). The top and the two bottom 1D-SAXS profiles are measured using an in-house SAXS setup. (**c**) 2D-SAXS scattering pattern for the F-duplex at 247 mg ml$^{-1}$. (**d**) $G_{20T}$-duplex samples in two-phase coexistence observed between cross-polarizers (total concentrations from left to right: 195.3, 203.5, 219.5 and 231.8 mg ml$^{-1}$). The red bars indicate the height of the isotropic region. (**e**) 2D-SAXS pattern for a shear-aligned sample of the $G_{20T}$-duplex at 242.0 mg ml$^{-1}$, (**f**) for the $G_{1T}$-duplex at 238.5 mg ml$^{-1}$ and (**g**) for a magnetically aligned sample of the $G_{20T}$-duplex at 255.0 mg ml$^{-1}$. The 2D-SAXS images presented in (**c**,**e**–**g**) are taken with a Pilatus 1 and 2 M detector, respectively; the red colour corresponding to the highest intensity. The red and black dotted lines are a guide for the concentration dependence of the $q_{DNA}$ and $q^*$ peaks, respectively.

Finally, Fig. 2b (second panel from the bottom) also presents the 1D-SAXS profile taken from the birefringent region of a $G_{20T}$-duplex solution (219.5 mg ml$^{-1}$) that exhibits two-phase coexistence (Fig. 2d). The data clearly indicate that the birefringent region is a nematic phase, and since smectic ordering is already present at a DNA concentration of 231.8 mg ml$^{-1}$, we conclude that the concentration range for a single-phase nematic in solutions of $G_{20T}$-duplexes is rather narrow.

**Phase diagram of the F- and G-duplex.** The information extracted from SAXS experiments on samples with many different concentrations, visual inspection of the samples between cross-polarizers, as well as optical textures observed by POM which will be discussed below, allow us to map out the phase diagram for the $G_{20T}$-duplex and F-duplex solutions as a function of the total DNA concentration up to 300 mg ml$^{-1}$.

The phase diagrams are given in Fig. 3. The F-duplex exhibits a $I/N^*$ coexistence region between 195 and 215 mg ml$^{-1}$. The chiral character of the nematic phase within and above the biphasic region is illustrated in the left inset of Fig. 3. This POM image, obtained by controlled drying experiments (more details can be found in the 'Methods' section), exhibits the typical cholesteric fingerprint texture with pitch of $P \sim 2.15 \, \mu m$, in line with previous experiments on duplexes with a similar contour length[31]. The POM image also indicates the presence of isolated dislocations within the cholesteric stripe structure (indicated by white arrows in the left inset in Fig. 3). The biphasic region is much narrower than observed for slightly polydisperse rigid dsDNA fragments. In particular, for dsDNA with $N_{bp} \sim 146 \pm 12$,

the width of the biphasic region was found to be between 135 and 271 mg ml$^{-1}$ (ref. 17), demonstrating that our synthetic F-duplex is extremely well-defined in length. The absence of a smectic-type of ordering up to 300 mg ml$^{-1}$ is therefore 'not' due to polydispersity in length.

The phase diagram of the $G_{20T}$-duplex is given in the lower panel of Fig. 3. Similarly to the F-duplex, the birefringent phase within the biphasic region and in a quite narrow window of concentrations above the biphasic region (indicated by the green region in the phase diagram in Fig. 3), exhibits a cholesteric fingerprint texture as illustrated in the lower second from the left image of Fig. 3. However, the pitch of $P = 1.15 \, \mu m$ is about half that of the $N^*$-phase for the F-duplex. In addition, a typical fan texture with many disclinations superimposed on the fingerprint bands is observed on a larger length scale within the narrow $N^*$-region (the second from the left, upper image). Such textures are characteristic for cholesteric phases, but also for smectic and columnar liquid crystalline phases. For higher concentrations (the blue region in the phase diagram in Fig. 3), the fan texture remains but the chiral pitch fingerprint bands are absent (third image from the left), in accordance with the existence of a $Sm$-A phase as revealed by the SAXS experiments.

The last POM image displayed in Fig. 3 (far right inset) is particularly significant. This was taken on the same magnetically aligned 255 mg ml$^{-1}$ solution of $G_{20T}$-duplexes used in the SAXS measurement described above, which showed smectic layer peaks (Fig. 2g). The optical texture shows an array of parabolic focal conics (PFCs), which are well-known defects characteristic of a smectic-A layer structure in thermotropic LCs[35]. The parabolic lines in the image are paired: one parabola lies in the plane of the

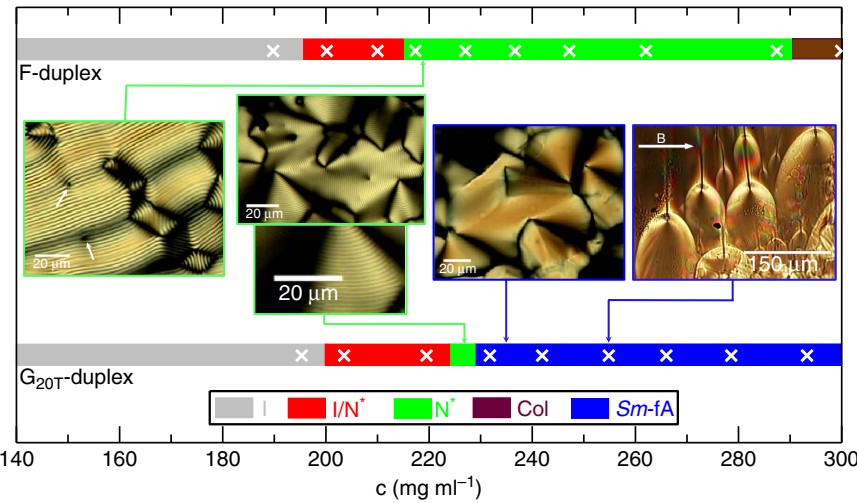

**Figure 3 | Phase diagram of gapped versus full-paired DNA duplexes.** Phase diagrams at room temperature for the $G_{20T}$–duplex in the bottom panel, and for the full-paired F-duplex in the top panel, as a function of the total DNA concentration. The colour code for the different phases is given in the lower right insert. The white crosses indicate the concentrations of samples that were loaded into capillaries for SAXS measurements. Phase identification was carried out by the combination of SAXS experiments, visual inspection of the capillaries between cross-polarizers and selective recording of the optical textures as observed by POM. POM images of DNA samples, confined in flat capillaries with thicknesses varying between 20 and 50 μm, are presented as inserts with coloured arrows that indicate their location in the phase diagram. The second from the left two POM images are for the same concentration of the $G_{20T}$–duplex, where the lower image is a magnification of the lower left part of the upper image, showing the chiral nature of the $N^*$-phase. The white arrows in the most left POM image indicate the presence of isolated dislocations within the cholesteric stripe texture. The most right POM image of a magnetic-aligned 255 mg ml$^{-1}$ solution of $G_{20T}$-duplexes, taken from a thin sample area located at the walls of a round X-ray capillary.

image and the other shares the same axis but lies in an orthogonal plane. The pairs thus appear as wishbones in the image. The vertex of each parabola in a pair passes through the focus of the other parabola in the pair. The parabolic lines are loci of conical cusps in the distorted smectic layer structure. PFCs are produced when the layers are strained along the layer normal, due either to a bulk stress or a stress associated with anchoring conditions at a boundary surface. The orientation of the PFC axes perpendicular to the magnetic field direction in the image is consistent with an equilibrium-layer-normal being perpendicular to the field, and hence with the negative diamagnetic susceptibility anisotropy for the DNA duplexes[33].

The observation of PFCs in the $G_{20T}$ system further confirms smectic layering. In fact, their dimensions scale as expected with those observed in classical small molecule smectic LCs. Namely, the ratio of layer spacings between the two systems is $\sim 11:1$, about the same as the ratio of spatial separation between the foci in a PFC pair ($\sim 15$–$25$ μm in the $G_{20T}$ smectic versus typically $\sim 1.3$–$2.9$ μm in the small molecule smectic according to ref. 35).

**Packing of $G_{20T}$-duplexes inside the smectic phase.** SAXS experiments on the $G_{20T}$-duplex revealed a lamellar structure with an average spacing between adjacent layers of $d \sim 34$ nm. Additional information concerning the arrangement of the $G_{20T}$-duplexes in the layers can be provided from the electron density profile. The latter can be extracted from the experimental X-ray scattering intensity and used to calculate the thickness of the DNA layer. Such an electron density profile $\rho_e(z)$ along the direction perpendicular to the layer plane $z$ is given in the lower panel of Fig. 4 (details of the method used in obtaining the electron density profile are given in Supplementary Fig. 2 and Supplementary Note 2). As expected most of the scattering comes from layers composed by dsDNA segments with average thickness of 31 nm, separated by a 3 nm layer mostly composed of ssDNA and water. The layer spacing is close to one molecular length ($L$) if one considers that the flexible part (ssDNA) is almost

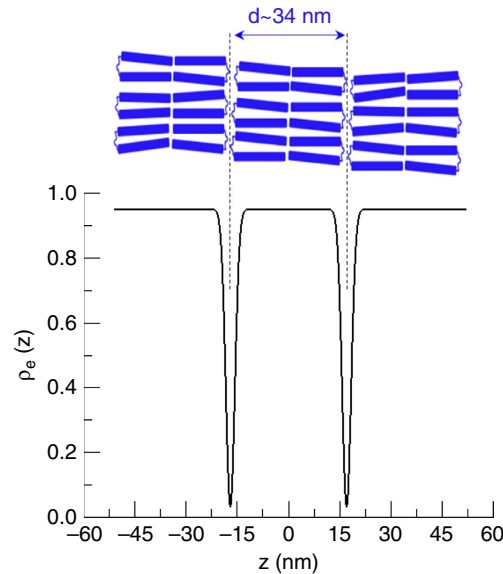

**Figure 4 | Electron density profile reconstruction.** Schematic of the arrangement of the gapped DNA molecules inside the smectic phase together with the calculated electron density profile extracted from the SAXS profile of $G_{20T}$-duplex at a concentration of 242 mg ml$^{-1}$ (see Fig. 2b).

collapsed ($L = 2 \cdot L_{dsDNA} + L_{ssDNA} = 2 \cdot 0.33 \cdot N_{bp} + L_{ssDNA} = 31.7$ nm $+ L_{ssDNA}$, with the contour length of ssDNA of 20 T bases equals to 12.6 nm, assuming the length of each base close to 0.63 nm (ref. 30)). Altogether with computer simulations discussed in the next section, this leads us to propose the molecular arrangement depicted schematically in the top panel of Fig. 4. In this packing scenario, the two stiff parts of the $G_{20T}$-duplex are folded and then stacked to form two halves of a layer, with the flexible parts, in a collapsed state, occupying the space

between layers. Such a molecular ordering also explains why the $G_{1T}$-duplex does not form a smectic phase, since folding is energetically unfavourable as compared with the much longer and more flexible spacer in the $G_{1T}$-duplex.

**Monte Carlo simulations.** To gain a deeper insight into the smectic ordering observed in gapped DNA solutions, we have carried out Monte Carlo (MC) simulations. The $G_{20T}$-duplexes are modelled in a coarse-grained manner as two hard cylinders with length $L = 16$ nm and thickness $D = 3$ nm (aspect ratio $X_0 = L/D = 5.33$). Each cylinder is decorated with two interacting sites, designated A and B. Referring to Fig. 5a, site B is the centre of the orange sphere (diameter $\sigma$), while site A is the centre of the small green sphere (diameter $\delta$) at the opposite end of the red cylinder. Site B is located along the symmetry axes at a distance $L/2 + \sigma/2$ from the centre of mass of the cylinder. The interaction potential $u_P$ between sites B is taken as 0 if $r < \sigma$ and $\infty$ otherwise, where $r$ is the distance between the sites. The interaction range $\sigma$ (that is, the diameter of the sphere associated to attractive sites B) in our simulations is taken equal to half of the contour length (12.6 nm) of the flexible 20T-spacer; this length has been estimated, assuming the length of each base, to be 0.63 nm (ref. 30). If the two cylinders belong to two distinct gapped duplexes, the interaction potential between their sites B is 0 for each $r$. Site A is located on the symmetry axis of the cylinder at a distance equal to $L/2 + 0.15D/2$ from the cylinder's centre of mass, and sites A belonging to two distinct cylinders interact via a square well potential $\beta u_{SW} = \beta u_0$ if $r < \delta$ and $\beta u_{SW} = 0$ if $r > \delta$, where $\delta = 0.25D$ is the interaction range (that is, the diameter of the sphere associated to interacting sites A).

The choices for the geometry and interaction potential of the B sites ensure full flexibility of our G-duplex without any energetic cost associated with bending it. The diameter $D$ of the stiff parts of the duplex is chosen to be larger than the steric diameter of DNA, which is around 2 nm, to account for electrostatic repulsion. Our choice of $D = 3$ nm is based on the effective diameter estimates reported in refs 36,37, using a salt concentration equal to 100 mM and a DNA concentration around 200 mg ml$^{-1}$, which amounts to an equivalent 800 mM salt concentration. The A sites account for hydrophobic interactions between the terminals of the duplexes[38], and their geometry is the same as the one used in ref. 39. The attraction strength between the hydrophobic patches is set to $\beta u_0 = 8.06$. The resulting stacking free energy is in line with values previously determined from the phase behaviour[16,39,40] and cholesteric properties[41] of self-assembling ultrashort DNA duplexes. More details regarding the simulation are given in the 'Methods', Supplementary Fig. 3 and Supplementary Method 2.

The phase behaviour and molecular organization of the simulated $G_{20T}$-duplexes is studied by calculating the equation-of-state, the fraction of folded G-duplex $\eta_f$, the order parameter, the three-dimensional (3D) pair-distribution function $g(\mathbf{r})$ (ref. 16), and by visual inspection of configurations (snapshots of selected phases). We define a folding fraction $\eta_f = \langle N_f^{45} \rangle / N$, where $\langle N_f^{45} \rangle$ is the average number of G-duplexes whose symmetry axes form a folding angle $\theta_f < 45°$ ($\theta_f = 0°$ corresponds to fully folded), and $N$ is the total number of particles.

The simulated equation-of-state is shown in Fig. 5b, where the dimensionless pressure $\beta P v_0$ ($v_0$ = volume of a single cylinder) is plotted against the DNA concentration. The simulations reveal a first-order transition from $I$ to a liquid crystalline state, as clearly

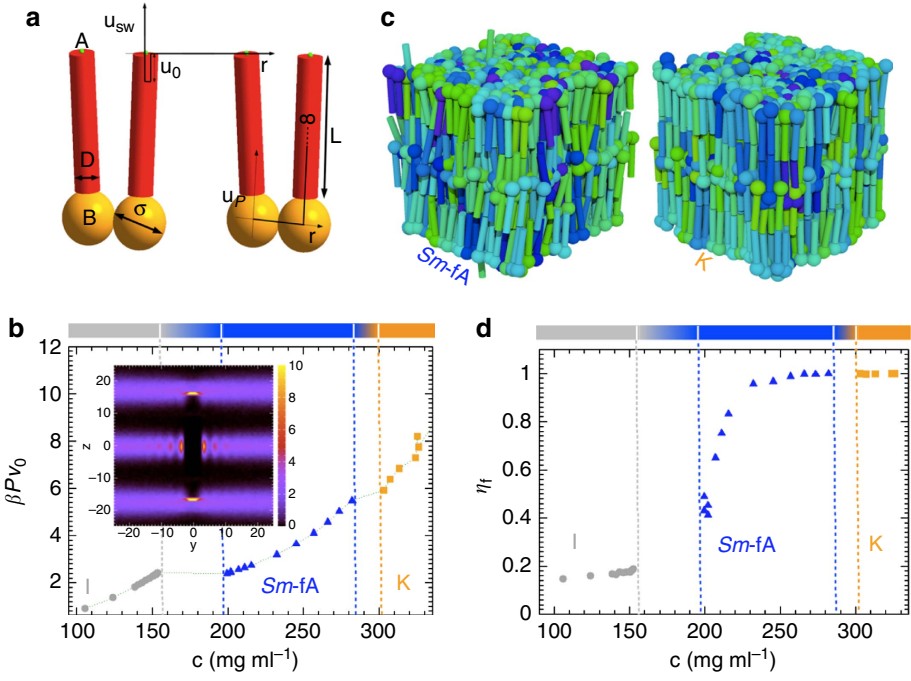

**Figure 5 | Monte Carlo simulations. (a)** The model for the $G_{20T}$-duplex molecule used in the simulations. The red parts are hard-core, stiff cylinders. The centres A of the small green spheres, which model the end-to-end attraction, interact via the square well potential $u_{SW}$ (shown on the top), while the centres B of the big orange sphere belonging to the same gapped duplex, which model the flexible spacer, interact via the potential $u_P$ (shown on the bottom-right). The diameter of the orange and green sphere indicate the interaction range and $\mu_0$ is the depth of the well of the square well potential $\mu_{SW}$, that is, it is the binding energy. **(b)** Equation-of-state for the $G_{20T}$-duplex obtained from simulations. Inset: plot of $g(0, y, z)$, which corresponds to correlations parallel to the nematic director ($z$-axis), for the $Sm$-fA state point corresponding to $\beta P v_0 = 4.1$. **(c)** Snapshots of selected phases. Cylinders belonging to the same molecule have the same colour to evidence the overwhelming number of duplexes in folded configurations. Left: the $Sm$-fA phase for $\beta P v_0 = 4.1$. Right: the crystal K-phase for $\beta P v_0 = 6.4$. **(d)** Fraction of folded G-duplex $\eta_f$ as a function of concentration.

indicated by the marked break in the $\beta P v_0$ versus concentration curve. The ordered phase exhibits a layered structure perpendicular to the nematic director (which is directed along $z$) as evidenced by the pair-distribution function $g(0, y, z)$ shown as an inset of Fig. 5b. We can thus unambiguously identify this phase with a smectic-A type LC.

The appearance of the smectic phase instead of a nematic phase just above the biphasic coexistence region is consistent with the very narrow concentration range where a full nematic state is found in the experiments (see Fig. 3). The absence of a nematic phase in the simulation can be understood in terms of an overestimate of the G-duplex flexibility in the simulation. Further compression of the smectic phase leads to a crystal ($K$) phase for concentrations above $300\,\mathrm{mg\,ml}^{-1}$ (which is beyond the concentration range where experiments have been performed). The quantitative characterization of these phases, based on the calculation of pair-distribution functions and the fraction of folded duplexes is discussed in the Supplementary Note 3 (see also Supplementary Figs 4,5 and 7–9).

Snapshots of the above mentioned two mesophases are depicted in Fig. 5c. The snapshot in the left panel reveals the Sm-fA molecular arrangement, where most duplexes are folded and where the flexible parts accumulate between the layers, similar to the cartoon in Fig. 4. Folding in the simulations can be quantified by computing the parameter $\eta_f$, which is shown in Fig. 5d as a function of concentration. It can be seen that the fraction of folded gapped duplexes at the $I$ to Sm-A transition abruptly changes from $\approx 0.15$–$0.2$, which corresponds to a uniform distribution of angles, to $0.4$–$0.5$, which signals a significant fraction of folded duplexes. The onset of the smectic-fA phase can therefore identified with the discontinuous jump of the fraction of folded gapped duplexes to values higher than those for a uniform distribution (more information regarding the angular distributions $P(\theta)$ of the gapped duplexes at different pressures $\beta P v_0$, is presented in Supplementary Fig. 6 and Supplementary Note 3). On further increasing the concentration, $\eta_f$ continuously increases until it reaches the value 1 in the $K$-phase.

We note that folding of particles leads to a significant reduction of excluded volume between G-duplexes, thus providing an effective way to minimize system free energy. A numerical estimate of the excluded volume in the smectic phase for fully unfolded ($v_{excl}$) and fully folded ($v'_{excl}$) yields $v_{excl}/v'_{excl} \cong 1.4$ (see Supplementary Note 4 for more details).

The weak blunt-end attractions that are typical for DNA are essential for the formation of the smectic-fA phase. Without these attractions we do find a phase with single layers of folded duplexes. These single layers, however, have a limited extent and do not regularly stack like in the smectic-fA phase (see Supplementary Fig. 10 and Supplementary Note 5). The blunt-end attractions are therefore necessary to obtain a regular layer stacking.

The sharp peak found in the SAXS experiments (Fig. 2b, top 1D-SAXS profile) for higher DNA concentrations indicates strong positional correlations between the G-duplexes within the smectic layers. This peak may possibly be associated with the formation of a smectic-B phase, wherein the G-duplexes are organized on a crystal lattice within the layers, but further investigation is needed to establish this conjecture. Simulations show that at even higher concentrations in the smectic phase, the folded G-duplexes remain isotropically arranged within the layers (see the 3D pair-distribution function $g(x, y, 0)$ in the Supplementary Fig. 7 and Supplementary Note 3 for further relevant discussion). The simple model for a gapped duplex assumed in our simulations probably needs further refinements in order accurately predict the occurrence of more highly ordered phases, such as smectic-B and columnar phases.

## Discussion

Smectic ordering in suspensions of gapped DNA duplexes is unambiguously demonstrated by SAXS experiments, in combination with the examination of sample textures by POM. MC simulations suggest that the DNA duplexes attain a predominantly folded conformation in the smectic phase. The incorporation of a sufficiently long, flexible ssDNA spacer in the middle of the stiff dsDNA rod-like molecule evidently leads to the stabilization of the lyotropic smectic phase that is not present for the stiff, fully paired dsDNA analogue.

The absence of smectic ordering in solutions of stiff, monodisperse (synthetic) F-duplexes is a clear manifestation of the crucial role of the attractive stacking interaction[14] between the duplexes' blunt-terminal ends. This attraction also implies that the unfolded conformation of gapped duplexes would inhibit smectic-type ordering: In an unfolded state, the presence of blunt-end enthalpic DNA interactions would induce the formation of a polydisperse set of linear semi-flexible aggregates[14,16], which would frustrate packing of the system into uniform smectic layers. On the other hand, an almost fully folded conformation of $G_{20T}$-duplexes allows for a 'self-protection' of the attractive DNA terminal sites, thus suppressing the formation of polydisperse linear aggregates and consequently accommodating a uniform layer structure.

The simulations we performed support the key experimental observations, despite the simplicity of the assumed model and afford some insight into the physical mechanism which leads to the formation of a smectic-fA phase. To stabilize the smectic phase, one does not need a fully folded system, but just a fraction of folded gapped duplexes sufficient to inhibit linear aggregation, that is, the formation of a set of polydisperse linear aggregates in the system[16]. Indeed, in our simulations the onset of the smectic-fA phase coincides with the presence of a fraction of folded duplexes significantly higher than in the isotropic phase (see Supplementary Fig. 6 and Supplementary Note 3).

In addition, the simulations again highlight the important role played by blunt-end hydrophobic interactions between the stiff DNA parts. Without these attractive attractions, the simulations predict folding of the duplexes in single layers of a limited size and random orientation (details are given in Supplementary Fig. 10 and Supplementary Note 5). The folding by itself to form such a micro-phase separated phase is thus of a purely entropic nature. The blunt-end interactions that are typical for DNA are necessary to align and order the single layers of limited size to form a smectic-fA phase.

The entropic forces originate from the flexibility mismatch between the covalently connected but chemical similar dsDNA and ssDNA segments, and act to segregate the stiff from the flexible blocks of the G-duplex. This scenario, based on a phase separation mechanism, is in line with predictions of Flory's mean-field theory on the phase behaviour of mixed solutions of rod-like particles and random polymer coils[42] and with experimental reports on entropically driven phase separation in mixtures of solutes which are sufficiently dissimilar in flexibility (such as rods and polymer coils[43,44], self-assembled filaments with different flexibility[45] and dsDNA and ssDNA short fragments[46]), or which differ significantly in length and/or diameters (like bidipserse rods[47]) or in persistence length (such as DNA in a suspension of nematic $f$d-virus[48]).

It is also worth mentioning, that existing theoretical and simulations studies of self-assembly in purely steric model systems[49–53], each consisting of stiff and flexible blocks, predict that the introduction of flexibility could possibly stabilize the smectic-A phase at the expense of nematic. However, in the systems considered, the flexible block is a terminal tail attached to a stiff rod, whereas in our gapped DNA system, flexibility is introduced locally within the DNA rod. Moreover, to our best knowledge, the above referenced model systems do not have a

true experimental equivalent, since it is a challenge to construct a system without introducing Flory–Huggins-type repulsive interactions due to the different chemical nature of the blocks.

Additional experiments, involving G-duplexes which are terminated with short non-sticky PolyT overhangs could be an interesting future direction to investigate further the crucial role of end-to-end enthalpic DNA interactions in the stabilization of the proposed smectic-fA phase. Such a modification in ultrashort DNA duplexes is known to create a steric hindrance at their terminal ends, and hence to suppress the end-to-end adhesion[14].

From the peculiar features of the chiral nematic phase of the $G_{20T}$-duplex, one may speculate that interesting analogies could emerge between the chiral nematics formed by DNA-based chain-sticks and the twist-bend nematic type of self-organization[54–56] that was recently found for achiral molecular dimers[57]. It would be intriguing to investigate the possibility of a twist-bend nematic in DNA chain-sticks, perhaps by utilizing shorter spacers.

Gapped DNA duplexes with various architectures represent a new class of lyotropic LC materials with a rich self-assembly behaviour, and one in which complex-structured phases are formed that do not exist in other types of materials (such as the smectic-fA phase described in our present work).This system is particularly attractive because the position and length of the stiff and flexible blocks can be chosen at will and controlled with a sub-nanometre precision. The unique physicochemical properties of DNA thus offer ways to engineer complex-architected molecules solely made of DNA and to tune the interplay between entropic and enthalpic interactions.

## Methods

**Synthesis of F-duplex and G-duplex.** Custom oligonucleotides were purchased from Biomers (www.biomers.net) and purified by high-performance liquid chromatography. The DNA concentration was determined by measuring the absorbance at 260 nm with a micro-volume spectrometer (NanoDrop 2000). Each DNA duplex was assembled by mixing a stoichiometric quantity of the strands involved in the gapped and fully paired duplex in 1 × TE/Na buffer (10 mM Tris, pH 7.5, 0.1 mM EDTA, 150 mM NaCl). The final concentration was 10 μM for each strand. The oligo mixtures were cooled slowly from 90 °C to room temperature in 10 l water placed in a styrofoam box over 48 h to facilitate strand hybridization. In all, 10% non-denaturing PAGE gels (Biorad) run in 1 × TBE (pH 8.3, Tris-borate-EDTA) buffer were used to confirm the assembly of each duplex. The electrophoresis experiment presented in Fig. 1b was performed on the crude reactions. The desired DNA structures migrate as a single sharp band, suggesting that F-duplex and G-duplexes were properly formed. More details regarding the DNA sequences used for the assembly of F-duplexes and G-duplexes are given in Supplementary Method 1.

**Sample preparation.** The samples were step-like diluted with buffer solution (10 mM Tris, pH 8.0, 150 mM NaCl) from highly concentrated solutions. The highest DNA concentration was prepared using a SpeedVac concentrator (Eppendorf). In every dilution step the DNA solution was thoroughly homogenized (up to 3 days for the more viscous samples) ensuring the absence of spatial concentrations gradients before loading into capillaries for SAXS experiments. The highest DNA concentration, achieved with our 'bulk' sample preparation method, was close to 300 mg ml$^{-1}$.

**Small-angle X-ray scattering.** Synchrotron-based SAXS (S-SAXS) measurements were performed at the Dutch-Belgian Beamline (DUBBLE) station BM26B (ref. 58) of the European Synchrotron Radiation Facility in Grenoble (France) and on the beamline 7.3.3 of the Advanced Light Source at Lawrence Berkeley Laboratory in USA. The in-house SAXS (H-SAXS) measurements were performed on the high brilliance Galium Anode Low Angle X-ray Instrument (GALAXI) of the Jülich Center for Neutron Science (JCNS, Germany). A Dectris-Pilatus 1M detector with resolution of 981 × 1,043 pixels and a pixel size of 172 × 172 μm² used to record the 2D-SAXS scattering patterns from H-SAXS (Jülich) and S-SAXS (DUBBLE). For the S-SAXS (Berkeley) measurements, a Pilatus 2M detector (1,475 × 1,679 pixels) was used. The 2D-SAXS patterns were integrated using FIT2D software. DNA solutions were loaded into 2-mm thickness borosilicate X-ray capillaries (Hilgenberg). The capillaries were sealed and stored at 4 °C for at least 1 month before used for X-ray experiments. Long-term stability and reproducibility was confirmed by repeating SAXS measurements on selected samples almost 1 year later.

**POM/Controlled drying experiments.** The polarized microscopy images presented in Fig. 3 were recorded on a coloured CMOS camera (Motic) which is installed in a Axioplan 2 upright microscope working in transmission mode between crossed polarizers. The phase behaviour of the $G_{20T}$-duplex and F-duplex was examined by controlled drying experiments in which concentrated DNA solutions in the isotropic phase were loaded by a capillary action into hollow rectangle glass tubes (VitroCom), with tube thickness varying between 20 and 50 μm, and sealed only from one side. The direction of the evaporation was thereby controlled, resulting in an increasing concentration gradient of DNA (and salt) across the tube. Images analysis was performed with Image J.

**MC simulations.** We carried out MC simulations in the constant pressure ensemble of $N = 840$ G-duplexes, using a cluster-NPT algorithm adapted from the one proposed in ref. 59 to speed up the equilibration process. In our MC simulations the box is allowed to change its size independently along the three directions $xyz$ and we use periodic boundary conditions. All quantities calculated from simulations are obtained by averaging during a production run of at least $5 \times 10^6$ MC steps, carried out after a proper equilibration stage during which we check the thermodynamic properties of the systems, such as internal energy, pair-distribution function and so on. The initial configuration for the equilibration run is generated as a crystalline lattice of fully folded parallel cylinders in an almost cubic lattice as discussed in refs 60,61 (see Supplementary Fig. 3 and Supplementary Method for more details).

To further address the thermodynamic stability of the smectic-fA phase, we carried out MC simulations starting with a broad distribution of folding angles corresponding to a fully equilibrated isotropic phase. Although in this simulation a fully equilibrated final state is not achieved within the very long simulation time span, due to the slowness of folding kinetics, we find clear evidence of a partially folded state, with only about 20% of fully unfolded duplexes remaining and with nematic order parameter $S \approx 0.5$, which shows a clear trend towards greater values as the system evolves toward full equilibration. The simulations are described in more detail in Supplementary Note 6 (see also Supplementary Fig. 11).

**Data availability.** The data that support the findings of this study are available from the corresponding authors on request.

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

## Acknowledgements

NWO is acknowledged for providing beamtime at the ESRF. G.P. and E.S. thank the BM26 staff, in particular Daniel Hermida-Merino, for their technical support during the beamtime. E.K. and E.S. thank Ulrich Rücker for experimental assistance with the in-house SAXS setup (GALAXI). M.S., J.T.G., A.J. and S.S. thank the National Science Foundation for supporting this research under grant no. DMR13-07674; M.S. also acknowledges support from an Institute for Complex Adaptive Matter (ICAM) Post-doctoral Fellowship. M.S., C.Z., J.T.G., A.J. and S.S. are grateful to the scientific staff at beamline 7.3.3 of the ALS. The Advanced Light Source is supported by the Director, Office of Science, Office of Basic Energy Sciences, of the U.S. Department of Energy under Contract No. DE-AC02-05CH11231. C.D.M. gratefully acknowledges support from PRIN-MIUR 2010-11 and thanks to Prof. Francesco Sciortino for useful discussions.

## Author contributions

E.S. conceived the study and designed research; J.Z. and E.S. performed synthesis; M.S., J.Z., G.P., J.T.G., A.J., C.D.M., J.K.G.D., S.S. and E.S. performed research; G.P. operated the DUBBLE S-SAXS beamline; C.Z. operated the Berkeley S-SAXS beamline; E.K. operated the Jülich H-SAXS beamline; M.S., G.P., J.K.G.D., S.S. and E.S. analysed experimental data; C.D.M. performed the Monte Carlo simulations and analysed numerical results; and C.D.M., J.K.G.D., S.S. and E.S. wrote the paper. All the authors made contributions to the writing of the manuscript and approved the final version.

## Additional information

**Competing financial interests:** The authors declare no competing financial interests.

**Publisher's note**: 

