## [Peer Review File · Nature Communications]

Reviewers' comments:

Reviewer #1 (Remarks to the Author):

This manuscript describes the first observation of a smectic phase in concentrated suspensions of "gapped" DNA. The experimental evidence for smectic A order in this system is unambiguous, and this work makes a significant contribution to our basic understanding of the self-assembly of nucleic acid-based soft matter. However, the proposed "folded" smectic A (smectic-fA) structural model seems implausible, and is not adequately supported by the presented experimental and simulation results, as discussed in more detail below. For this reason, I cannot recommend publication of this article in Nature Communications in its present form. However, a substantially revised manuscript may be suitable for publication.

As far as I can tell from the manuscript, there is no direct or indirect experimental evidence for the proposed smectic-fA structural model. In fact, the x-ray scattering results are consistent with a continuum of possible models, ranging from a fully unfolded model, with each gapped duplex spanning the entire 34 nm smectic layer, to the fully folded model proposed by the authors. It's highly probable that the entropy of mixing of folded and unfolded molecules leads to a situation intermediate between these two extremes. Moreover, the conformational free energy of folded molecules is expected to be higher than that of unfolded molecules, which tends to disfavor thermodynamic states with a preponderance of folded gapped duplexes. In light of this, the fully folded smectic-fA model put forward by the authors seems rather implausible.

The only evidence for the smectic-fA structural model presented in the manuscript comes from Monte Carlo simulations of a coarse-grained molecular model of gapped DNA duplexes, but even here the evidence seems mixed, as the fraction of folded duplexes ranges from ~ 0.4 to ~ 1 with increasing concentration within the smectic-fA phase, as shown in Figure 5d. Also, as far as I can tell, all simulations were started in a fully folded configuration, so it's possible that the large fraction of folded molecules observed at high concentrations is an artifact arising from the folded initial condition and slow kinetics of interconversion between folded and unfolded molecular states. Further simulations starting with a fully unfolded initial condition would be quite useful in this regard, as would special-purpose Monte Carlo moves to interconvert folded and unfolded duplexes. Finally, the authors state that the folding of particles leads to a significant reduction of excluded volume between G-duplexes (page 6, last sentence in the second full paragraph). Further discussion of this point would be useful, as it is far from obvious why this is the case, nor is it clear what specific features of the simulation model are responsible for this reduction in excluded volume.

Other comments:

The description of the simulation model is rather confusing and lacks important details. The ss-DNA chain linking the two duplexes (represented as hard cylinders) is represented by two "big orange patches" whose interaction energy is zero if they overlap and infinite otherwise, but as far as I can tell the authors don't say where these patches are centered; from the figures, I'd guess that each patch is centered on a point on the corresponding cylinder axis at a distance $\sigma/2$ from the cylinder end, but the reader shouldn't have to guess. A better schematic illustrating the model would be very helpful

here. Also, the term "big orange patch" is confusing; why not just specify the interaction potential between the two interaction sites, wherever they're located? Similarly, the attractive interaction between duplex blunt ends isn't fully specified; it seems that this is a square well interaction between cylinder endpoints, but the range of the attraction isn't specified, although it can be inferred to some degree from Figure 5. Finally, saying that the interaction is between "green spheres" is confusing; it'd be better to simply say that the interaction is between interaction sites located at the center of the cylinder faces, I think.

The authors suggest that the occurrence of a smectic phase in gapped DNA is unexpected. However, there is an extensive body of prior theoretical and simulation work that shows that the introduction of flexibility in mesogenic molecular models can lead to entropic stabilization of smectic phases at the expense of the nematic phase [see, for example: A. Casey and P. Harrowell, *J. Chem. Phys.* 110, 12183 (1999); J. S. van Duijneveldt, A. Gil-Villegas, G. Jackson, and M. P. Allen, *J. Chem. Phys.* 112, 9092 (2000); C. McBride and C. Vega, *J. Chem. Phys.* 117, 10370 (2002); D. Duches and D. E. Sullivan, *J. Phys.: Condens. Matter* 14, 12189 (2002); R. C. Hidalgo, D. E. Sullivan, and J. Z. Y. Chen, *J. Phys.: Condens. Matter* 19, 376107 (2007)]. The authors should discuss their results in the context of this prior work.

Reviewer #2 (Remarks to the Author):

I recommend publishing without revision.

The appearance of a smectic phase in suspensions of gapped DNA duplexes is surprising. The three dsDNA mesogens described in this paper - ungapped, gapped with 1 ssDNA nucleotide and gapped with 20 ssDNA nucleotide are sufficient to demonstrate that the extra flexibility is required for smectic formation.

I liked the use of simulation to elucidate the mechanism of the smectic phase formation. They found single layers of folded duplexes of limited extent and random orientation with gapped duplexes without attraction. But when attractions between the end of the duplexes was included, the smectic bilayer phase observed in experiment was replicated in the simulations.

In experiments, the appearance of a nematic and columnar phase and the lack of a smectic phase in the F-duplex case are all consistent with end-to-end association of the F-duplex to create a polydisperse system that suppresses the smectic. Is there any experimental evidence to suggest that end-to-end association is what suppresses the smectic in the F-duplex?

REVIEWERS' COMMENTS:

Reviewer #1 (Remarks to the Author):

The authors have responded thoughtfully to the concerns raised in my earlier review, and the result is an improved manuscript, both in terms of clarity and content. I recommend acceptance of this manuscript for publication in Nature Communications without further revision.

Reply to the comments of Referees

We would like to thank both referees for their very useful and constructive remarks, which give us the opportunity to revise and to improve our manuscript. Our responses to specific points are presented in the following way: We first quote a comment from the referees' reports (in blue), then provide our specific reply, followed by details of the relevant revisions made to the manuscript. Besides what follows, we have made several corrections of typos, and have done some editorial polishing to make the manuscript read more smoothly.

Referee #1

We appreciate the referee's view that our work represents "the first observation of a smectic phase in concentrated suspensions of "gapped" DNA" and that it "makes a significant contribution to our basic understanding of the self-assembly of nucleic acid-based soft matter".

While the referee acknowledges that "the experimental evidence [presented] for smectic A order in this system is unambiguous", he/she expresses critical concerns about the plausibility of our model for the smectic layer structure based on a folded conformation of the gapped duplexes. Below we will focus primarily on these concerns, which are certainly valid to raise. Overall, in the revised manuscript and SI we have made an effort to stress that our model is a *proposal* that is supported indirectly by experiment, and more directly by simulations (including notable, new results described below) and more detailed arguments (e.g., an excluded volume comparison).

A. Response to main comments of Referee 1

Referee's Comment: As far as I can tell from the manuscript, there is no direct or indirect experimental evidence for the proposed smectic-fA structural model. In fact, the x-ray scattering results are consistent with a continuum of possible models, ranging from a fully unfolded model, with each gapped duplex spanning the entire 34 nm smectic layer, to the fully folded model proposed by the authors. It's highly probable that the entropy of mixing of folded and unfolded molecules leads to a situation intermediate between these two extremes. Moreover, the conformational free energy of folded molecules is expected to be higher than that of unfolded molecules, which tends to disfavor thermodynamic states with a preponderance of folded gapped duplexes. In light of this, the fully folded smectic-fA model put forward by the authors seems rather implausible.

Our reply: We agree with the referee that it is not possible from the X-ray scattering measurements to extract *direct* evidence for the self-folding of gapped duplexes within the smectic layers. The extracted layer spacer could be compatible with either a folded or fully unfolded packing scenario. However, there are compelling experimental results and interpretation that disfavor the unfolded packing scenario – particularly, the *absence* of a smectic-type of ordering in the fully-paired duplex (*F*-duplex), which cannot fold, and the

reasons behind it. The main arguments to explain the absence of a smectic are based on strand flexibility, length polydispersity and the recently discovered attractive stacking interaction between the terminal ends of dsDNA. The latter induces linear aggregation, resulting in a polydisperse rod-like system that suppresses smectic layering.

In our manuscript, we investigated the LC behavior of a *synthetic*, fully-paired, stiff dsDNA fragment, which has a fixed number of base pairs. Our *F*-duplex, with length $L_{dsDNA} = 116$ bp and ratio w of persistence length (l_p^{dsDNA}) over contour length (L_{dsDNA}) close to 1.3 showed no indication of smectic ordering (see Fig. 1a and Fig. 3 of our original manuscript).

[redacted]

Regarding the issue of the conformational free energy of folded molecules being higher than the unfolded state, we note that the only entropic cost consequent to folding of our system is the stretching of the flexible ssDNA spacer. Since the persistence length of the flexible polymeric spacer (ssDNA) is of the order of 3 bases, and in total the spacer length is 20 bases, the entropic cost associated with folding is in fact quite small.

In response to the referee's comment and considering the points in our reply above, we have revised our manuscript as follows:

Page 8, second column, second paragraph of the Discussion, we have added a new paragraph: *The absence of smectic ordering in solutions of stiff, monodisperse (synthetic) F-duplexes is a clear manifestation of the crucial role of the attractive stacking interaction [14] between the duplexes' blunt-terminal ends. This attraction also implies that the unfolded conformation of gapped duplexes would inhibit smectic-type ordering: In an unfolded state, the presence of blunt-end enthalpic DNA interactions induces the formation of a polydisperse set of linear semi-flexible aggregates [14, 16], which would frustrate packing of the system into uniform smectic layers. On the other hand, the almost fully folded conformation of G_{20T}-duplexes allows for a "self-protection" of the attractive DNA terminal sites, thus suppressing the formation of polydisperse linear aggregates and consequently accommodating a uniform layer structure.*

Referee's comment: The only evidence for the smectic-fA structural model presented in the manuscript comes from Monte Carlo simulations of a coarse-grained molecular model of gapped DNA duplexes, but even here the evidence seems mixed, as the fraction of folded duplexes ranges from ~0.4 to ~1 with increasing concentration within the smectic-fA phase, as shown in Figure 5d.

Our reply: We apologize for not providing a clear definition of smectic-fA phase in our manuscript. A smectic-fA can be defined as a smectic-A phase where the fraction of folded gapped duplexes is significantly higher than in the isotropic phase, and thus not necessarily equal to 100%. Indeed, to stabilize the smectic phase one does not need a fully folded system but just a fraction of folded gapped duplexes sufficient to inhibit linear aggregation, i.e. the formation of a set of polydisperse linear aggregates in the system. To clarify this point, Figure 3 (below) shows the distribution of folding angles for several Isotropic and Sm-fA state points. It can be seen that a fraction of unfolded duplexes is still present in the system in the smectic phase.

Figure 3. Distribution of the angle between the two cylinders belonging to the same gapped duplex for different pressures for isotropic ($\beta P v_0 = 0.91, 1.37$ and 1.82), and smectic states ($\beta P v_0 = 2.74, 3.19$ and 3.65).

Regarding this point, we have revised our manuscript – specifically, the Introduction, Monte Carlo simulations section, Discussion section, and the Supplementary Information – as follows:

Page 2, first column, second sentence from the end of the Introduction section, the sentence is changed to: *Based on a combination of physical arguments and our simulation results, we propose a specific model for the smectic layer structure in which the gapped duplexes predominantly adopt a folded configuration, with the rigid parts of our DNA-based chain-sticks lie side by side.*

Page 8, end of first column, the following sentences are added: *The onset of the smectic-fA can therefore be identified with the discontinuous jump of the fraction of folded gapped duplexes to values higher than those for a uniform distribution. (More information on the angular distribution of the folding angle at different pressures $\beta P v_0$, which shows a jump in the population of folded duplexes relative to a uniform distribution, is presented in SI, section 2.1, Supplementary Fig. S6).*

Page 9, first column, 2nd paragraph from the top, the following text is added: *...and afford some insight into the physical mechanism that leads to the formation of a smectic-fA phase. In order to stabilize the smectic phase, one does not need a fully folded system, but just a fraction of*

folded gapped duplexes sufficient to inhibit and limit the formation of polydisperse linear aggregates in the system [16]. Indeed, in our simulations the onset of the smectic-fA phase coincides with the presence of a fraction of folded duplexes significantly higher than in the isotropic phase. (see SI, section 2.2, Supplementary Fig. S6).

Supplementary Information, bottom of page S6, the following sentences are added: *The distribution of folding angles for several isotropic and smectic state points is shown in Fig. S6. It can be seen that in the Sm-fA phase the overwhelming number of duplexes attain a folded configuration, and this number increases on increasing the pressure.*

Supplementary Information, top of page S8, Figure 3 presented above has been added as a new Supplementary Figure (S6).

Referee's comment: Also, as far as I can tell, all simulations were started in a fully folded configuration, so it's possible that the large fraction of folded molecules observed at high concentrations is an artifact arising from the folded initial condition and slow kinetics of interconversion between folded and unfolded molecular states. Further simulations starting with a fully unfolded initial condition would be quite useful in this regard, as would special-purpose Monte Carlo moves to interconvert folded and unfolded duplexes.

Our reply: We agree on the importance and usefulness of the additional simulations suggested. In fact, we have conducted and recently obtained preliminary results from one such simulation, which we believe strongly support the folded conformation in the smectic state. Specifically, we have analyzed recent MC simulations that are started from an equilibrated distribution of folding angles (angle between two cylinders of the same duplex) in the *isotropic* phase, with the same value of $\beta u_0 = 8.06$ as in the simulations reported in our original manuscript and for two different pressures. Although full equilibration is not achieved over a very long simulation timespan (about 3 months), due to the slowness of folding kinetics, we do find clear evidence of a partially folded mesophase (containing only about 20% of fully unfolded duplexes) with nematic order parameter $S \sim 0.5$. As demonstrated in Fig. 4b (below), the order parameter, plotted as a function of MC steps, also shows a clear trend towards higher values on further equilibrating the system. Moreover, as shown in Fig. 4a, the distribution of the folding angle (calculated by averaging over the last 5×10^6 MC steps of the equilibration runs), bears a remarkable resemblance to the distribution obtained at equilibrium for the Sm-fA phase from the simulations we reported in the manuscript. These preliminary results make us confident that Sm-fA phase is thermodynamically stable.

We would also like to point out that when computer simulating phase transitions from a less ordered to a more ordered phase, it is a common practice to start from a state with some degree of order to speed up the otherwise very long equilibration process. For example, in the papers by

D. Frenkel, *J. Phys. Chem.* **92**, 3280-3284 (1988) and S. C. McGrother *et al.*, *J. Chem. Phys.* **104**, 6755 (1996), cited in our first submission, the phase diagram of concentrated hard core particles has been studied through isobaric Monte Carlo simulations. A smectic phase for aspect ratio $X_0=L/D=5$ is obtained starting from orientationally-ordered (nematic) configurations. The slow kinetics of formation of an ordered state starting from a less ordered initial configuration in such systems is discussed, for example, in Kuriabova *et al.*, *J. Mat. Chem.* **20**, 10366-10383 (2010).

Figure 4. (a) Distribution of the angle between the two cylinders belonging to the same gapped duplex for different pressures. The two curves labeled as “compression” have been obtained from equilibration runs with a hydrophobic attraction $\beta v_0 = 8.06$ starting from an initial isotropic configuration at $\beta P v_0 = 3.2$ and 4.1. (b) Order parameter as a function of MC steps for

the two-equilibration runs at pressures $\beta P v_0 = 3.2$ and 4.1 . The straight line is just a guide to the eye, which evidences the ongoing equilibration process.

Therefore, we have added in our manuscript clarifying remarks along the lines of our reply to the referee in the **Methods section and Supplementary Information**.

Page 10, end of the Monte Carlo Simulation section in Methods, the following text is added: *To further address the thermodynamic stability of the smectic- fA phase, we carried out MC simulations starting with a broad distribution of folding angles corresponding to a fully equilibrated isotropic phase. Although in this simulation a fully equilibrated final state is not achieved within the very long simulation time span, due to the slowness of folding kinetics, we find clear evidence of a partially folded state, with only about 20% of fully unfolded duplexes remaining and with nematic order parameter $S \approx 0.5$, which shows a clear trend towards greater values as the system evolves toward full equilibration. The simulations are described in more detail section 2.4 of SI.*

Supplementary Information, page S12, a new paragraph with title “*Equilibration runs from the isotropic phase*” is added together with Fig. 4 from our reply above.

Because the promising new simulation results are still preliminary, we have changed language throughout our manuscript to remind the reader that simulations “suggest”, rather than “reveal”, the folded scenario for the packing of gapped DNA to form smectic layers. We have also edited the text to avoid language that might imply that the experimental results at their current stage could provide a *direct* evidence of the folded model.

Referee’s comment: Finally, the authors state that the folding of particles leads to a significant reduction of excluded volume between G-duplexes (page 6, last sentence in the second full paragraph). Further discussion of this point would be useful, as it is far from obvious why this is the case, nor is it clear what specific features of the simulation model are responsible for this reduction in excluded volume.

Our reply: In the smectic A phase with the nematic director along the z-axis, the excluded volume of two anisotropic particles can be written as follows (e.g. see *Wessels, P. P. F. and Mulder, B. M. Soft Mat. Nematic Homopolymers: From Segmented to Wormlike Chains. Soft Mat. 1, 313-342 (2003)*)

$$v_{excl} = -\frac{1}{V} \int d\mathbf{R}_1 d\mathbf{R}_2 d\Omega_1 d\Omega_2 e_{12}(\mathbf{R}_{12}, \Omega_1, \Omega_2) f_l(z_1, \Omega_1) f_{l'}(z_2, \Omega_2)$$

where $\mathbf{R}_1 = (x_1, y_1, z_1)$ and $\mathbf{R}_2 = (x_2, y_2, z_2)$ are the positions of the two particles, $\mathbf{R}_{12} = \mathbf{R}_1 - \mathbf{R}_2$, Ω_1 and Ω_2 are their orientations and

$$e_{12}(\mathbf{R}_{12}, \Omega_1, \Omega_2) = \exp[-U_h(\mathbf{R}_{12}, \Omega_1, \Omega_2)/k_B T] - 1$$

is the Mayer function with U_h being the interaction potential between the two particles.

For two gapped duplexes either in a fully unfolded or fully folded configuration we calculated numerically the excluded volume v_{excl} and v'_{excl} respectively by using a Monte Carlo integration. The numerical procedure is identical to the one discussed in *De Michele, C. et al., Macromolecules* **45**, 1090-1116 (2012), except that in the present case the centers of mass of the two gapped duplexes are constrained to be onto the plane $z=0$ to account in a simplified way for the smectic layering in the system. From our numerical calculation we obtain that: $v_{excl}/v'_{excl} \approx 1.4$.

We have included this result in our revised manuscript as follows:

Page 8, second column, second paragraph from the top, the following text was added: *A numerical estimate of the excluded volume in the smectic phase for fully unfolded (v'_{excl}) and fully folded (v_{excl}) yields $v_{excl}/v'_{excl} \approx 1.4$ (see SI, section 2.2 for more details).*

In addition, a new paragraph with title “*Excluded volume calculations in the Smectic phase*” was added in the **Supplementary Information, page 10**. This paragraph contains the above-mentioned reply to referee together with an extra citation.

B. Response to additional comments of Referee 1:

Referee’s comment: The description of the simulation model is rather confusing and lacks important details. The ss-DNA chain linking the two duplexes (represented as hard cylinders) is represented by two "big orange patches" whose interaction energy is zero if they overlap and infinite otherwise, but as far as I can tell the authors don't say where these patches are centered; from the figures, I'd guess that each patch is centered on a point on the corresponding cylinder axis at a distance $\sigma/2$ from the cylinder end, but the reader shouldn't have to guess. A better schematic illustrating the model would be very helpful here.

Our reply: We apologize for having not provided all details of the model, which we used in the simulations. As observed by the referee, the center of the “big orange patches” lays along the axis of the cylinder at a distance $\sigma/2$ from the base. We completely rewrote the initial part of the section on Monte Carlo Simulation and we also changed Figure 5 and its caption accordingly (*marked with blue color in the resubmitted manuscript, Monte Carlo section*).

Referee’s comment: Also, the term "big orange patch" is confusing; why not just specify the interaction potential between the two interaction sites, wherever they're located? Similarly, the attractive interaction between duplex blunt ends isn't fully specified; it seems that this is a square

well interaction between cylinder endpoints, but the range of the attraction isn't specified, although it can be inferred to some degree from Figure 5.

Our reply: We agree with the referee and we apologize for not making this point clearer. We have explicitly added the definition of interaction potentials between the terminal sites in the revised text marked in blue on pages 6 and 7.

Referee's comment: Finally, saying that the interaction is between "green spheres" is confusing; it'd be better to simply say that the interaction is between interaction sites located at the center of the cylinder faces, I think.

Our reply: We agree with the referee on this point, and we reformulated text according to his/her suggestion (marked with blue color in the resubmitted manuscript, Monte Carlo section).

Referee's comment: The authors suggest that the occurrence of a smectic phase in gapped DNA is unexpected. However, there is an extensive body of prior theoretical and simulation work that shows that the introduction of flexibility in mesogenic molecular models can lead to entropic stabilization of smectic phases at the expense of the nematic phase [see, for example: A. Casey and P. Harrowell, *J. Chem. Phys.* 110, 12183 (1999); J. S. van Duijneveldt, A. Gil-Villegas, G. Jackson, and M. P. Allen, *J. Chem. Phys.* 112, 9092 (2000); C. McBride and C. Vega, *J. Chem. Phys.* 117, 10370 (2002); D. Duchs and D. E. Sullivan, *J. Phys.: Condens. Matter* 14, 12189 (2002); R. C. Hidalgo, D. E. Sullivan, and J. Z. Y. Chen, *J. Phys.: Condens. Matter* 19, 376107 (2007)]. The authors should discuss their results in the context of this prior work.

Our reply: We thank the referee for bringing these theoretical and simulations works to our attention, and we have added them to our revised manuscript. In addition, we have modified the *Discussion* section, in order to discuss our simulation work in the context of this prior work.

Although the addition of flexible tails to the ends of stiff rods can promote smectic order (as in the above references), it is not obvious that this would be the case if the flexible element is introduced in the middle of the rod. Indeed, as demonstrated for viral rods in *Pouget, E. et al, Phys. Rev. A* **84**, 041704 (2011), the nematic to smectic-A phase transition occurs at *higher* concentrations when the rod itself becomes more flexible. Of course, in our case the flexibility is not distributed isotropically along the DNA rod, but the situation is clearly distinct from the systems described in the references cited by the referee.

The specific revisions to our manuscript are:

Page 2, last paragraph of Introduction, 8th line from the bottom, the following reference was added in order to make clear what is the expected result of the effect of flexibility on the location

of the nematic-to-smectic transition: Pouget, E. et. al *Dynamics in the smectic phase of stiff viral rods*. *Phys. Rev. A* **84**, 041704 (2011).

Page 9, bottom of the first column, the following paragraph was added together with 5 new citations, as suggested by the referee: *It is also worth mentioning, that existing theoretical and simulations studies of self-assembly in purely steric model systems [49-53], each consisting of stiff and flexible blocks, predict that the introduction of flexibility could possibly stabilize the smectic-A phase at the expense of nematic. However, in the systems considered, the flexible block is a terminal tail attached to a stiff rod, whereas in our “gapped” DNA system, flexibility is introduced locally within the DNA rod. Moreover, to our best knowledge, the above referenced model systems do not have a true experimental equivalent, since it is a challenge to construct a real system without introducing Flory-Huggins-type repulsive interactions due to the different chemical nature of the blocks.*

Additionally, taking into account the points in the referee’s comment and our reply above, we have replaced the term “unexpected” with “unusual”, which we hope is a reasonable compromise.

Referee #2

We would like to thank the referee for his/her positive recommendation to accept our manuscript for publication.

Response to comments of Referee 2:

Referee’s comment: The appearance of a smectic phase in suspensions of gapped DNA duplexes is surprising. The three dsDNA mesogens described in this paper - ungapped, gapped with 1 ssDNA nucleotide and gapped with 20 ssDNA nucleotide are sufficient to demonstrate that the extra flexibility is required for smectic formation.

Our reply: We thank referee for reiterating this important experimental finding which was crucial for putting forward the folded packing scenario of gapped duplexes in the smectic phase

Referee’s comment: I liked the use of simulation to elucidate the mechanism of the smectic phase formation. They found single layers of folded duplexes of limited extent and random orientation with gapped duplexes without attraction. But when attractions between the end of the duplexes was included, the smectic bilayer phase observed in experiment was replicated in the simulations.

Our reply: We certainly agree on the importance of combining experiments and simulations in order to obtain a deeper level of understanding of the smectic-type of ordering in suspensions of DNA-based chain-sticks.

Referee's comment: In experiments, the appearance of a nematic and columnar phase and the lack of a smectic phase in the F-duplex case are all consistent with end-to-end association of the F-duplex to create a polydisperse system that suppresses the smectic. Is there any experimental evidence to suggest that end-to-end association is what suppresses the smectic in the F-duplex?

Our reply: As noted in the third paragraph of the *Introduction* section of our manuscript, direct experimental evidence for the presence of the attractive stacking interaction in DNA and its crucial role in self-assembly of short dsDNA fragments, was demonstrated in *Nakata et. Al, Science 318, 1276-1279 (2007)*, which we cite in Ref. 14. These authors showed that ultrashort blunt-ended dsDNA fragments (up to 20 bp synthetic dsDNA), with aspect ratio well below the value expected for the formation of any liquid crystal phase, do form a nematic phase above a critical concentration. A series of subsequent experimental (Ref. 15 in our manuscript) and theoretical/simulation (Ref. 16) works further established end-to-end association as an inherent property of dsDNA with terminal blunt-ends. So it is valid to assume that this attractive stacking interaction will be present also in the case of the F-duplex (dsDNA with blunt-ends) with length $L_{dsDNA} = 116$ bp.

It is also natural to hypothesize that, for rods with infinitely large persistence length, the impact of such end-to-end attraction in rod self-assembly will become weaker as the rod length increases, since the isotropic to liquid crystal transition is expected to appear at lower volume fractions. However, the validity of this hypothesis will break down for the DNA-based rod-like molecules, since their persistence length l_{dsDNA} is finite (~ 150 bp). In other words, an increase of the duplex length, resulting in increased flexibility, will suppress the formation of the smectic phase and even destabilize the nematic phase, as already demonstrated in Ref. 31 of our manuscript.

A possible experimental approach to test the above hypothesis concerning the impact of end-to-end attraction could be the construction of very long, stiff DNA rods with blunt-ends using the DNA origami approach (see, e.g., *Rothmund, P. W. K., Nature 440, 297-302 (2006)*; *Czogalla, A. et. Al, Nano Lett. 15, 649-655 (2015)*).